# A Machine Learning Method for Modeling Wind Farm Fatigue Load

**Yizhi Miao, Mohsen N. Soltani *** and **Amin Hajizadeh**

AAU Energy, Aalborg University, 6700 Esbjerg, Denmark; yimi@energy.aau.dk (Y.M.); aha@energy.aau.dk (A.H.)
* Correspondence: sms@energy.aau.dk

**Abstract:** Wake steering control can significantly improve the overall power production of wind farms. However, it also increases fatigue damage on downstream wind turbines. Therefore, optimizing fatigue loads in wake steering control has become a hot research topic. Accurately predicting farm fatigue loads has always been challenging. The current interpolation method for farm-level fatigue loads estimation is also known as the look-up table (LUT) method. However, the LUT method is less accurate because it is challenging to map the highly nonlinear characteristics of fatigue load. This paper proposes a machine-learning algorithm based on the Gaussian process (GP) to predict the farm-level fatigue load under yaw misalignment. Firstly, a series of simulations with yaw misalignment were designed to obtain the original load data, which considered the wake interaction between turbines. Secondly, the rainflow counting and Palmgren miner rules were introduced to transfer the original load to damage equivalent load. Finally, the GP model trained by inputs and outputs predicts the fatigue load. GP has more accurate predictions because it is suitable for mapping the nonlinear between fatigue load and yaw misalignment. The case study shows that compared to LUT, the accuracy of GP improves by 17% (*RMSE*) and 0.6% (*MAE*) at the blade root edgewise moment and 51.87% (*RMSE*) and 1.78% (*MAE*) at the blade root flapwise moment.

**Keywords:** machine learning; Gaussian process; damage equivalent load

## 1. Introduction

Wind farm owners schedule turbine layouts more and more tightly for higher profits. Unfortunately, this tight layout has a more pronounced wake effect, which leads to more power loss. Therefore, active wake control (AWC) is introduced to reduce the power loss caused by wake [1]. AWC can be implemented in two ways. One is wake steering control (WSC), which reduces the wake effect on the downstream turbine by deflecting the wake propagation direction through yaw misalignment [2]. The other is axial induced control (AIC), which reduces the wind energy capture of the upstream turbine by adjusting the generator speed and pitch angle [3]. The wake steering control has been proven to improve power production more than axial induction control in large eddy simulation (LES) simulation [4], wind tunnel [5], and field tests [6]. Moreover, the new research proved that WSC also affects the fatigue load on the downstream turbine, reducing turbine service life [7]. Consequently, recent developments in AWC have heightened the need for fatigue loads prediction.

Most studies have focused on fatigue load estimation at the turbine level, which did not consider the effects of yaw misalignment and wake interaction between turbines. Schröder et al. [8] proposed an artificial neural network (ANN) model to improve the prediction accuracy of fatigue load in a single turbine. Compared with the polynomial chaos expansion (PCE) model [9], the ANN model shows lower NRMS error, faster evaluation time, and small sample adaptability. However, the effect of wake features is not considered in this paper. Similar work such as Singh's [10] research uses chained Gaussian processes to provide probabilistic predictions, which is valuable for robust control [11,12].

Ervin Bossanyi [13] performed a study surrogate model for turbine loads in two-turbine farms. This surrogate model added different wind and control factors to the steady wind. The limitation of this paper is the accuracy of influence factors, which may differ in complex layouts. Georgios [14] applied the regression method to map the relationship between the load and de-rating control strategy. This method can estimate the short-term fatigue load. However, it does not consider the effect of yaw control. Finally, a multi-dimensional look-up table was proposed to predict the farm load in [15]. This method obtains the load prediction from interpolation, significantly reducing the computational time. However, with the increase in wind farms, this method will be time-consuming and inaccurate. Some studies have considered the balance between wind farm power generation and load [16,17]. Nevertheless, these studies rely on computational fluid dynamics models, which have unaffordable computing time.

In summary, the fatigue load prediction for wake steering control must have the following properties:

- Running fast: The fatigue load predictions will be used in the optimization loop. Therefore, the computational time must be as short as possible.
- Nonlinear mapping capability: The ambient wind and wake interaction influence the fatigue loads. Thus, a complex nonlinear relationship exists between fatigue load and ambient wind and wake interaction.
- Considering wake interaction: The load of the downstream turbine is influenced by the upstream turbine's wake effects, including partial wake, wake meandering, and wake superposition.
- Considering wake deflection: The wake deflection affects the load as it changes the wake propagation direction by yaw misalignment, so wake deflection must be considered.

Considering the above objectives, machine learning and artificial intelligence, such as SVM [18], GP [10], and DNN [19], are suitable for predicting the fatigue load because they can achieve the above goals at the same time.

This paper proposes a machine learning model based on GP to predict fatigue loads under yaw misalignment. Firstly, a series of mid-fidelity simulations based on FAST.Farm [20] with yaw misalignment were designed to obtain the original load data, which considered the wake interaction between turbines. Secondly, the rainflow counting and Palmgren miner rules were introduced to transfer the original load to damage equivalent load. Finally, the GP model predicts the fatigue load after training by inputs and outputs. The main contributions of this paper are as follows: (1) A mapping from ambient wind, wake interactions, and yaw misalignment to fatigue damage at farm level; (2) the effects of yaw misalignment on blade root edgewise moment and blade root flapwise moment were measured.

The remainder of this paper is organized as follows: Section 2 introduces the main framework and theoretical knowledge; the GP method and evaluation metrics are introduced in Section 3. Lastly, case study results and conclusions are presented in Sections 4 and 5, respectively.

## 2. Main Framework and Theoretical Knowledge

### 2.1. Main Framework

A typical machine learning process includes data collection, data preprocessing, input and output construction, model training, and testing. The load signal is affected by many aspects, such as wind speed (WS), wind direction (WD), turbulence intensity (TI), axial induction factor (de-rating), and yaw angles ($\theta$). The scope of this paper covers wind speed, wind direction, turbulence intensity, and yaw angles.

The flow chart of the main work is shown in Figure 1. In the beginning, the simulations are executed to obtain the original load data corresponding to the inputs. Then, the function of fatigue modeling is to calculate outputs from the original load data. Further, the GP model is trained by inputs and outputs. Finally, the fatigue load can be predicted by the trained GP model.

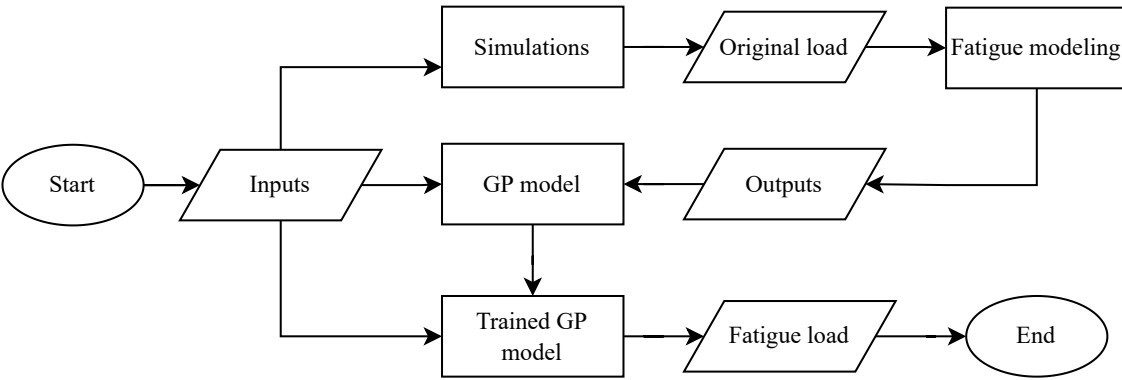

**Figure 1.** Fatigue load prediction flowchart.

### 2.2. Fatigue Load Modeling

The raw load data cannot visually show the equivalent damage value because it is presented in time series. Therefore the rainflow counting (RFC) method and the Palmgren–Miner (PM) rule were proposed to calculate damage equivalent load (DEL). The calculation process of DEL is shown in Figure 2. The RFC converts the load signals to a series of cycles ($n_i$) in terms of stress amplitude($s_i$) [21]. The PM rule combined with the S–N curve (material properties) transforms RFC results to equivalent damage load [22].

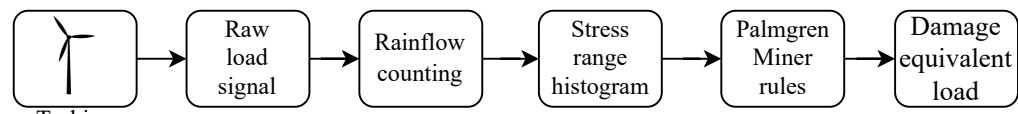

**Figure 2.** Damage equivalent load estimation procedure.

Suppose that the cycles $n_i$ and the corresponding stress amplitude $s_i$ are obtained by the rainflow counting algorithm. $i$ is calculated automatically and different $i$ represent different loop amplitudes. The ratio of fatigue cycles to total life failure cycles can be defined as damage factor $d_i$ [23]:

$$d_i = \frac{n_i}{N_i} \tag{1}$$

where $n_i$ is cycles calculated from RFC; $N_i$ is number of cycles to failure in terms of the stress amplitude $S_i$.

As shown in Figure 3, the PM rule is used to express the relationship between stress and the number of failure cycles, which is linear in the log–log coordinate system, where $m$ is the Wohler exponent.

The relationship can be expressed as

$$N_i = \left( \frac{S_0}{S_i} \right)^m \tag{2}$$

Then, Equation (1) can be expressed as

$$d_i = \frac{n_i}{N_i} = \frac{n_i S_i^m}{S_0^m} \tag{3}$$

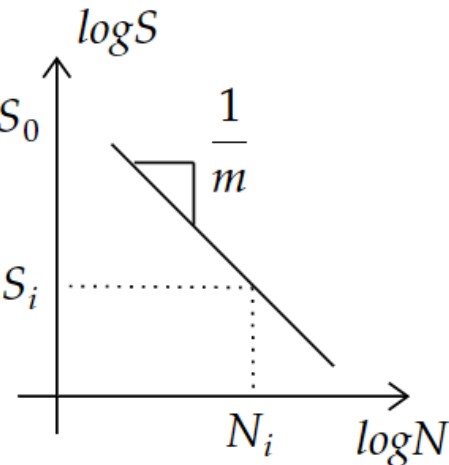

**Figure 3.** S–N curve : The relationship between the magnitude of alternating stress and the number of cycles to failure for a given material.

The total damage *DEL*(Damage Equivalent Load) can be summed for cycles of difference amplitudes:

$$DEL = \frac{1}{S_0^m} \sum n_i S_i^m \tag{4}$$

For a given number of cycles $n_{eq}$ and the corresponding amplitude $S_{eq}$, the DEL can be expressed as

$$\frac{n_{eq} S_{eq}^m}{S_0^m} = DEL = \frac{1}{S_0^m} \sum n_i S_i^m \tag{5}$$

Finally, when $n_{eq}$ equals signal length in seconds, the DEL of the signal is equal to the damage of a 1 Hz sinusoidal signal of the same length with amplitude $S_{eq}$. This method avoids the use of extreme loads.

$$S_{eq} = \left( \frac{n_i S_i^m}{n_{eq}} \right)^{\frac{1}{m}} \tag{6}$$

**3. Machine Learning Method**

*3.1. GP Model*

The GP is an excellent machine learning model with the distribution of functions [24]. It is a random process with good adaptability to complex problems such as small samples, nonlinearity, and high dimensionality. For a given dataset $D = \{(x_i, y_i)\}_{i=1}^n$, where $x_i \in R^d$ is the input data matrix, $y_i \in R$ is the output data matrix, given a limited set of data $D$, $f\left(x^{(1)}\right), f\left(x^{(2)}\right), f\left(x^{(3)}\right)$ can form a set of random variables and have a joint Gaussian distribution. All the statistical characteristics of the Gaussian process are defined by the mean function $m(x)$ and the covariance function $k(x, x')$.

$$f(x) \sim GP(m(x), k(x, x')) \tag{7}$$

*3.2. GP Model Prediction Procedure*

Figure 4 shows the GP prediction procedure, which consists of two phases, i.e., training and prediction. The training phase uses inputs and outputs to train the model, where the inputs include wind speed, wind direction, TI, and yaw angles, and the outputs are the damage equivalent loads from the fatigue load modeling. The predicted values are obtained if the inputs and the trained model are given. For a particular load on a wind farm with $n$ turbines, $n$ GP models are required.

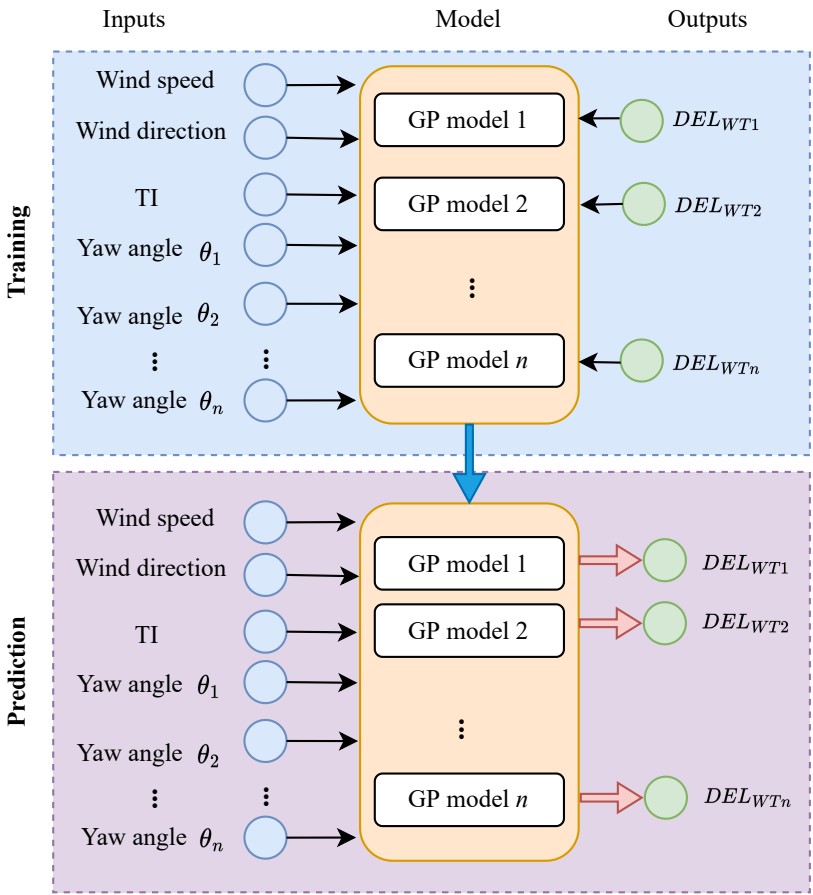

**Figure 4.** GP prediction procedure.

### 3.3. Evaluation Factor

The predicted fatigue loads will be used in the yaw offset optimization process. In addition, measuring the effect of yaw misalignment on the fatigue load will help decide whether the fatigue load needs to be considered. Thus, the following metrics are defined:

$$R_{WTi} = max(DEL_{WTi}(\theta_1, \theta_2, \ldots, \theta_n)) - min(DEL_{WTi}(\theta_1, \theta_2, \ldots, \theta_n)) \tag{8}$$

where $R_{WTi}$ is the max–min range of turbine $i$; $DEL_{WTi}(\theta_1, \theta_2, \ldots, \theta_n)$ is the the fatigue loads of turbine $i$ in the whole yaw angle cases.

The root-mean-square error (*RMSE*) and mean absolute error (*MAE*) are used to measure the accuracy of the prediction. *RMSE* and *MAE* are defined as follows [25]:

$$RMSE = \sqrt{\frac{1}{N} \sum_{i=1}^{N} (y_i - \hat{y}_i)^2} \tag{9}$$

$$MAE = \frac{1}{N} \sum_{i=1}^{N} |y_i - \hat{y}_i| \tag{10}$$

where $N$ is the number of evaluation points, and $y_i$ and $\hat{y}_i$ are the actual and predicted outputs. The smaller the *RMSE* and *MAE* are, the more accurate the prediction is.

## 4. Case Study

This section discusses the results of FAST.Farm simulations [20], damage equivalent loads, and the GP prediction. The parameters of simulations are shown in the first subsection.

As a mid-fidelity simulation, FAST.Farm has a satisfactory calculation accuracy than the low-fidelity simulation, such as FLORIS [26], and its computational cost is lower than the high-fidelity simulation, such as SOWFA [27]. FAST.Farm is based on an improved dynamic wake meandering (IDWM) model, which addresses many of the limitations of the original DWM model. The load calculation module of FAST.Farm relies on the ElastoDyn module in Openfast to calculate the structural load, including tower, nacelle, drivetrain, and blades.

### 4.1. Parameters

Wind farm simulation involves many parameters, including wind precursor, wind turbine and farm layout, and simulation control parameters.

1. Wind precursor: A precursor is the wind data used in a simulation to generate ambient wind. There are nine wind precursors with different wind speeds and turbulence intensities generated by TurbSim [28]. The average wind speed (hub level), wind direction, and turbulence intensity of precursors are shown in Table 1. A wind precursor case is shown in Figure 5. The U, V, and W components of the wind speed represent the horizontal, cross, and vertical directions of the wind speed in space, respectively.

2. Turbine and layout parameters: The NREL 5 MW wind turbine is used, with a diameter of 126 m and a nacelle height of 90 m [29]. The wind farm contains two or three wind turbines, which are placed on a straight line, with a distance of five times the diameter of the wind turbine. Figure 6 shows the velocity of FAST.Farm simulation at 1900 s under precursor P7 and yaw offset ($\theta_1 = 5$ and $\theta_2 = 0$).

3. Yaw angle control strategy: The control strategy mainly focuses on yaw-based control. Each simulation case runs at fixed yaw angles for 2000 s. The last 500 s of data are used to calculate DELs. The range of yaw angles command in two turbine cases is shown in Table 2. For example, in the control mode of Y1, the upstream wind turbine WT1 of each case will move to a fixed angle ($\theta_1$), and the downstream wind turbine will maintain an angle of 0 ($\theta_2 = 0$). Under Y2, the last turbine keeps zero yaw angles. Moreover, the load types include blade root edgewise moment (BREM) and blade root flapwise moment (BRFM), and their damage equivalent load values are DEL-BREM and DEL-BRFM.

**Table 1.** Parameters of wind precursor.

| Precursor | Wind Speed (m/s) | TI (%) | Wind Direction (Degree) |
|:---:|:---:|:---:|:---:|
| P1 | 5 | 5 | 0 |
| P2 | 5 | 10 | 0 |
| P3 | 5 | 15 | 0 |
| P4 | 7 | 5 | 0 |
| P5 | 7 | 10 | 0 |
| P6 | 7 | 15 | 0 |
| P7 | 9 | 5 | 0 |
| P8 | 9 | 10 | 0 |
| P9 | 9 | 15 | 0 |

**Table 2.** Parameters of yaw control.

| Mode | Yaw Angle of Turbine 1 (Degree) | Yaw Angle of Turbine 2 (Degree) | Yaw Angle of Turbine 3 (Degree) |
|---|---|---|---|
| Y1 | $\theta_1 \in [-25, 25]$ $\theta_1 \in \mathbb{N}$ [a] | $\theta_2 = 0$ | - |
| Y2 | $\theta_1 \in [-30, 30]$ $\theta_1 \in \mathbb{N}$ | $\theta_2 \in [-30, 30]$ $\theta_2 \in \mathbb{N}$ | $\theta_3 = 0$ |

[a] $\mathbb{N}$ is interger.

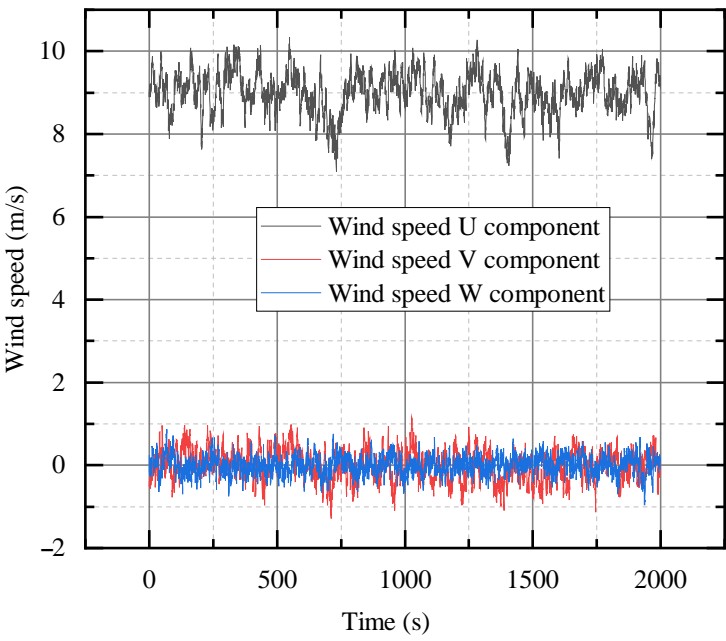

**Figure 5.** Precursor of wind speed at 9 m/s and TI at 5% (P7).

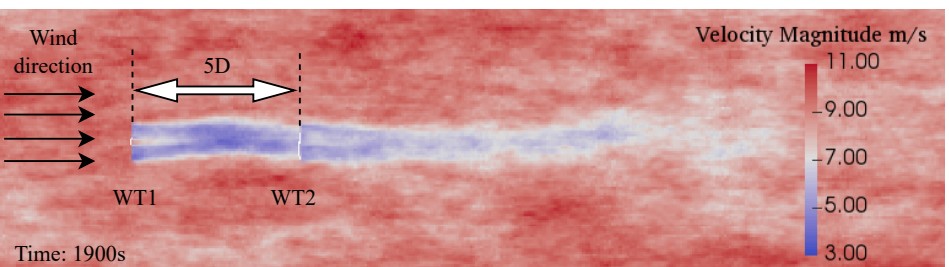

**Figure 6.** The 1900th second FAST.Farm simulation under precursor P7 and yaw offset ($\theta_1 = 5$ degree, $\theta_2 = 0$ degree).

*4.2. Fatigue Damage Results*

Figure 7 shows the original load signal of BREM and its 1 Hz equivalent signal (from fatigue damage modeling) under precursor P1. The 1 Hz equivalent signal for different loads has a fixed frequency and a different amplitude. The amplitude represents the value of the damage equivalent load. The figure shows that the 1 Hz equivalent signal has a lower amplitude than the original signal at higher frequencies.

Figure 8 shows the DEL-BREM calculation results of WT1 and WT2, including sample data and mean value. The sample data divide the original load into ten groups, and each group calculates the damage equivalent load through the fatigue load modeling.

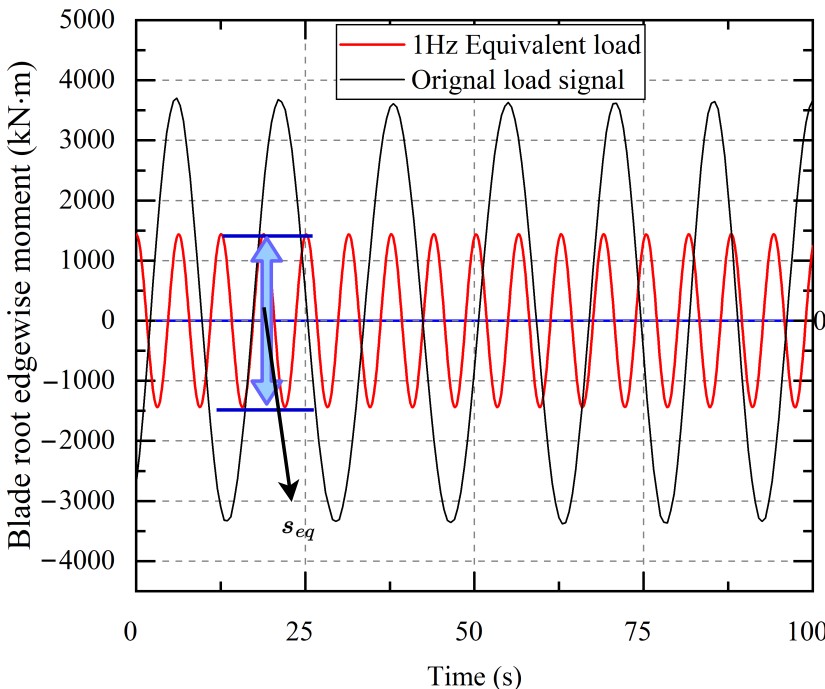

**Figure 7.** BREM orignal load signal and its 1 Hz equivalent signal under precursor P7.

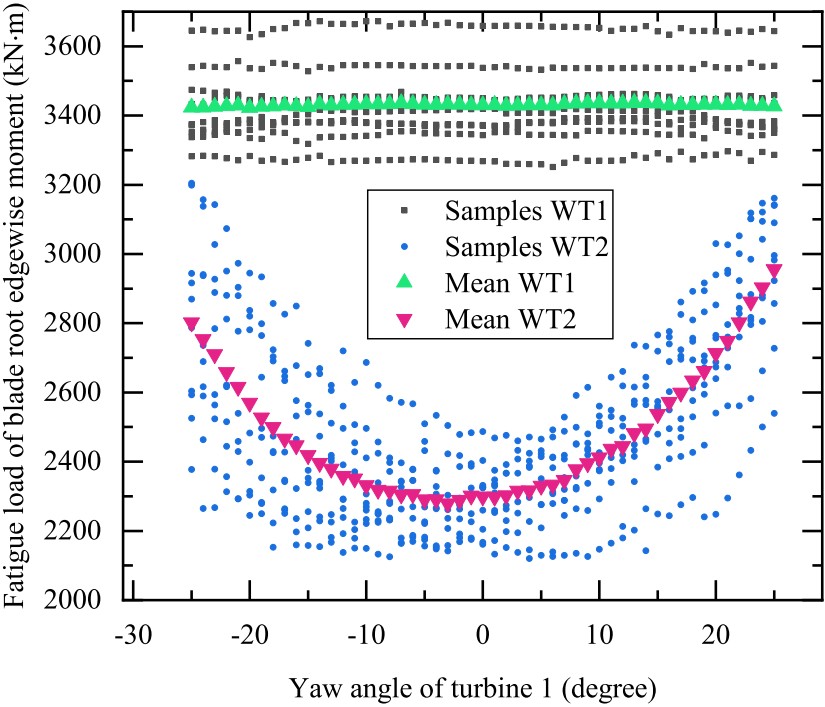

**Figure 8.** DEL-BREM's samples data and mean value in each yaw angle case.

*4.3. GP Prediction Results*

4.3.1. Two-Turbine Case

Figure 9 shows the predictions of LUT and GP in WT1 and WT2. In Figure 9a, the LUT prediction looks better than GP because the fatigue load of WT1 is close to a straight line. In contrast, the prediction of GP, Figure 9b, is more fitted to the samples than the LUT algorithm.

Figure 10 shows the 95% confidence level probabilistic prediction results from GP, which proves the predictions covered 9% of the samples.

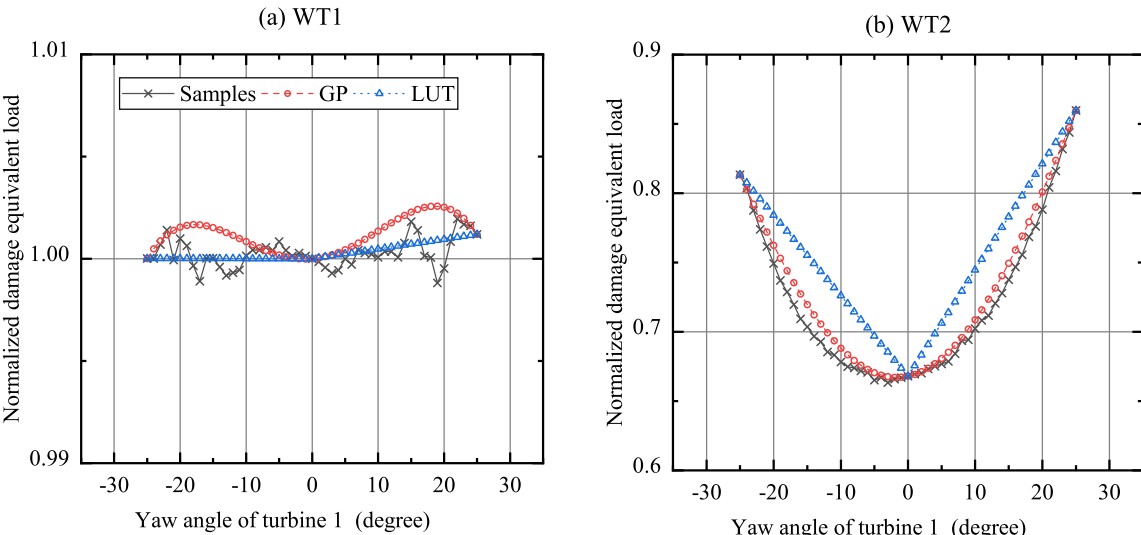

**Figure 9.** Prediction of normalized DEL-BREM by LUT and GP under yaw control mode Y1 and wind precursor P1.

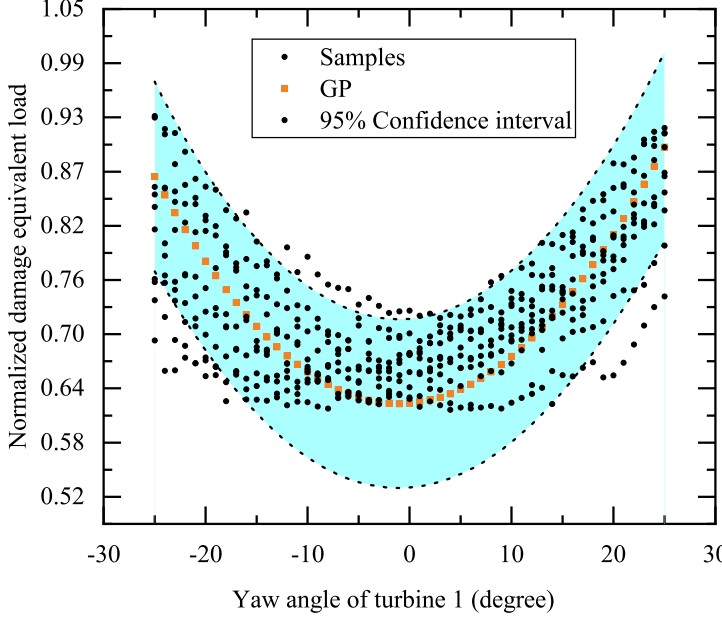

**Figure 10.** Probabilistic prediction of normalized DEL-BREM by GP under yaw control mode Y1 and wind precursor P1.

Tables 3 and 4 demonstrate the *RMSE* errors of BREM on WT2 for different wind speeds and turbulence intensities. Table 3 shows that the prediction errors of LUT and GP become more significant as the wind speed increases, from 22.69% to 50.09% for LUT and from 5.45% to 15.19% for GP. However, the prediction accuracy of GP increases with wind speed compared to LUT, from 17.24% to 34.90%. On the contrary, Table 4 shows that the prediction error decreases with the increase of turbulence intensity. The TI increases from 5% to 15%, the LUT prediction error decreases from 22.69% to 7.56%, and the GP decreases by about 2.5%. The improvement of GP relative to LUT prediction also decreases gradually.

In conclusion, the prediction error of GP increases with the wind speed and decreases with the increase of turbulence intensity. Similar conclusions can be obtained from Tables 5 and 6.

**Table 3.** *RMSE* error of blade root edgewise moment.

| DEL | Precursor | Wind Speed m/s | LUT RMSE/% | GP RMSE/% | Improvement RMSE/% |
|---|---|---|---|---|---|
| | P1 | 5 | 22.69 | 5.45 | 17.24 |
| BREM | P4 | 7 | 35.58 | 9.53 | 26.05 |
| | P7 | 9 | 50.09 | 15.19 | 34.90 |

**Table 4.** *RMSE* error of blade root edgewise moment.

| DEL | Precursor | TI /% | LUT RMSE/% | GP RMSE/% | Improvement RMSE/% |
|---|---|---|---|---|---|
| | P1 | 5% | 22.69 | 5.45 | 17.24 |
| BREM | P2 | 10% | 15.07 | 5.48 | 9.59 |
| | P3 | 15% | 7.56 | 2.95 | 4.61 |

**Table 5.** *RMSE* error of blade root flapwise moment.

| DEL | Precursor | Wind Speed m/s | LUT RMSE/% | GP RMSE/% | Improvement RMSE/% |
|---|---|---|---|---|---|
| | P1 | 5 | 48.88 | 9.21 | 39.67 |
| BREM | P4 | 7 | 164.16 | 43.13 | 121.03 |
| | P7 | 9 | 370.03 | 99.20 | 274.83 |

**Table 6.** *RMSE* error of blade root flapwise moment.

| DEL | Precursor | TI /% | LUT RMSE/% | GP RMSE/% | Improvement RMSE/% |
|---|---|---|---|---|---|
| | P1 | 5% | 48.88 | 9.21 | 39.67 |
| BREM | P2 | 10% | 36.88 | 10.36 | 26.52 |
| | P3 | 15% | 18.09 | 1.51 | 16.58 |

4.3.2. Three-Turbine Case

Figures 11 and 12 show the BREM and BRFM sample data on normalized damage equivalent loads in three turbines. For this case, the wind speed, turbulence intensity, and direction are 9 m/s, 5%, and 0 degrees. The axis scales of the four subplots are the same. The obtained fatigue load is normalized for visual expression by dividing a fixed value. The obtained fatigue load is normalized for visual expression by dividing a fixed value. The fixed values, in this case, take the fatigue load of the first turbine under the yaw offset [0, 0, 0].

**Remark 1.** *The first-row turbine's damage equivalent load in blade root edgewise moment is less affected by yaw misalignment, while the second- and third-row turbines are more affected by yaw misalignment.*

As shown in Figure 11, the normalized BREM-DELs of WT1 remain essentially constant, which proves that the BREM-DELs in the first row of turbines are less affected by yaw misalignment. In contrast, the yaw misalignment affects the BREM-DELs of the second and third rows of turbines because the BREM-DELs curves of WT2 and WT3 vary with yaw angles.

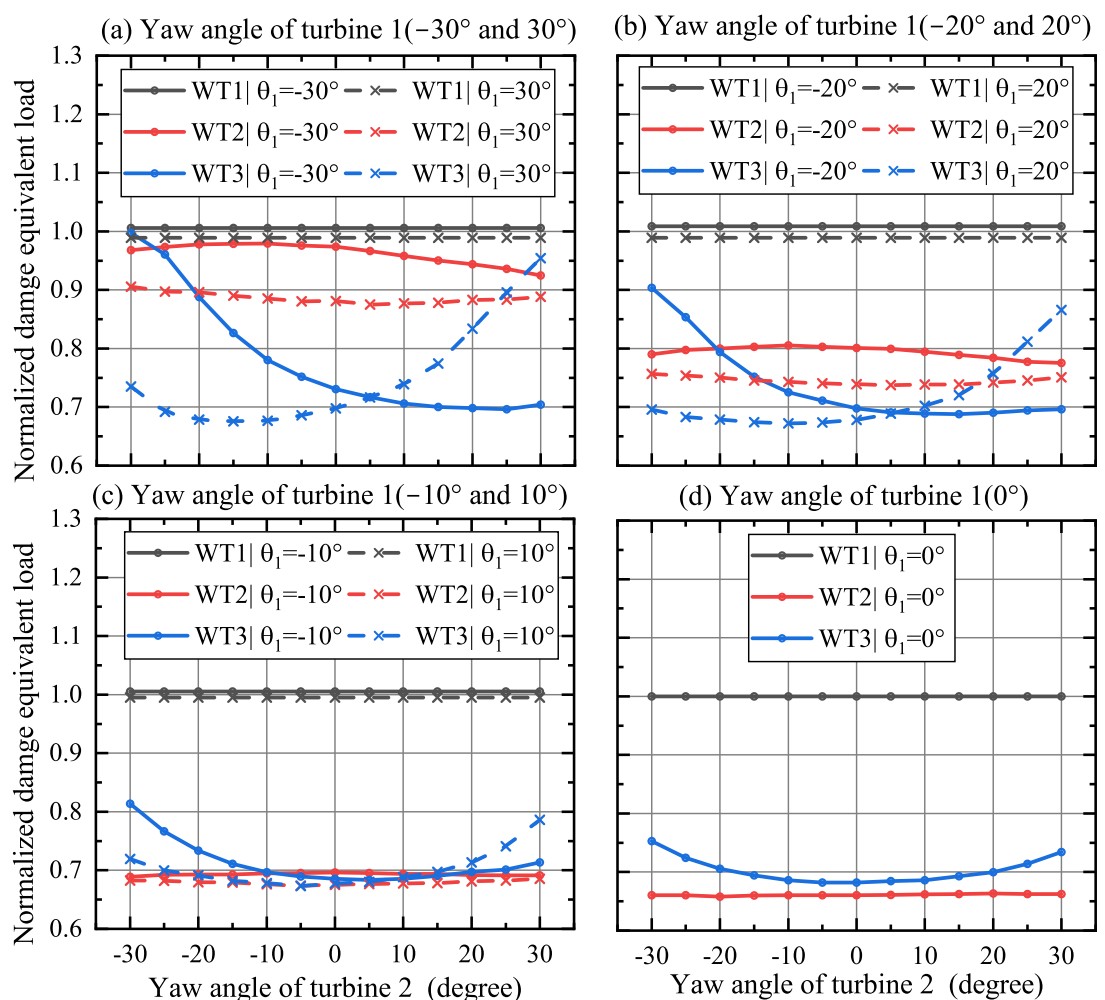

**Figure 11.** Blade root edgewise moment samples of normalized damage equivalent load under 9 m/s wind speed and 5% turbulence intensity. The damage equivalent load of turbine 1 under zero yaw offset case equals 1. The same axis scale is used in the four sub-pictures.

**Remark 2.** *The damage equivalent load in blade root flapwise moment is affected in all turbines.*

As shown in Figure 12, the normalized BRFM-DELs of WT1 stay constant when the yaw angle of WT2 changes. However, this changes with the yaw angle of WT1. Regarding WT2 and WT3, the normalized BRFM-DELs vary with WT1 and WT2 yaw angles. Although all three turbines are affected by yaw misalignment, the influencing factors differ. Specifically, the fatigue load of WT1 is affected by its yaw deflection. In addition, WT2 is affected by WT1 wake and its yaw deflection. WT3, on the other hand, is affected by the mixed wake of WT1 and WT2 since the yaw angle of WT3 is zero.

Figure 13 displays the max–min range (defined in Equation (8)) of normalized fatigue load . Figure 13a is blade root edgewise moment, and Figure 13b is blade root flapwise moment. In Figure 13a, the max–min range of WT1 is 2%, meaning the first row of turbines is almost unaffected by its yaw misalignment. On the other hand, the max–min range of WT2 and WT3 is 32% and 33% when the yaw misalignment is active. From Figure 13b, all turbines' blade root flapwise moments are affected by the yaw misalignment. The max–min ranges of WT1, WT2, and WT3 are 38%, 100%, and 75%, respectively. Thus, the above results confirmed the observations from Figures 11 and 12 (Remarks 1 and 2).

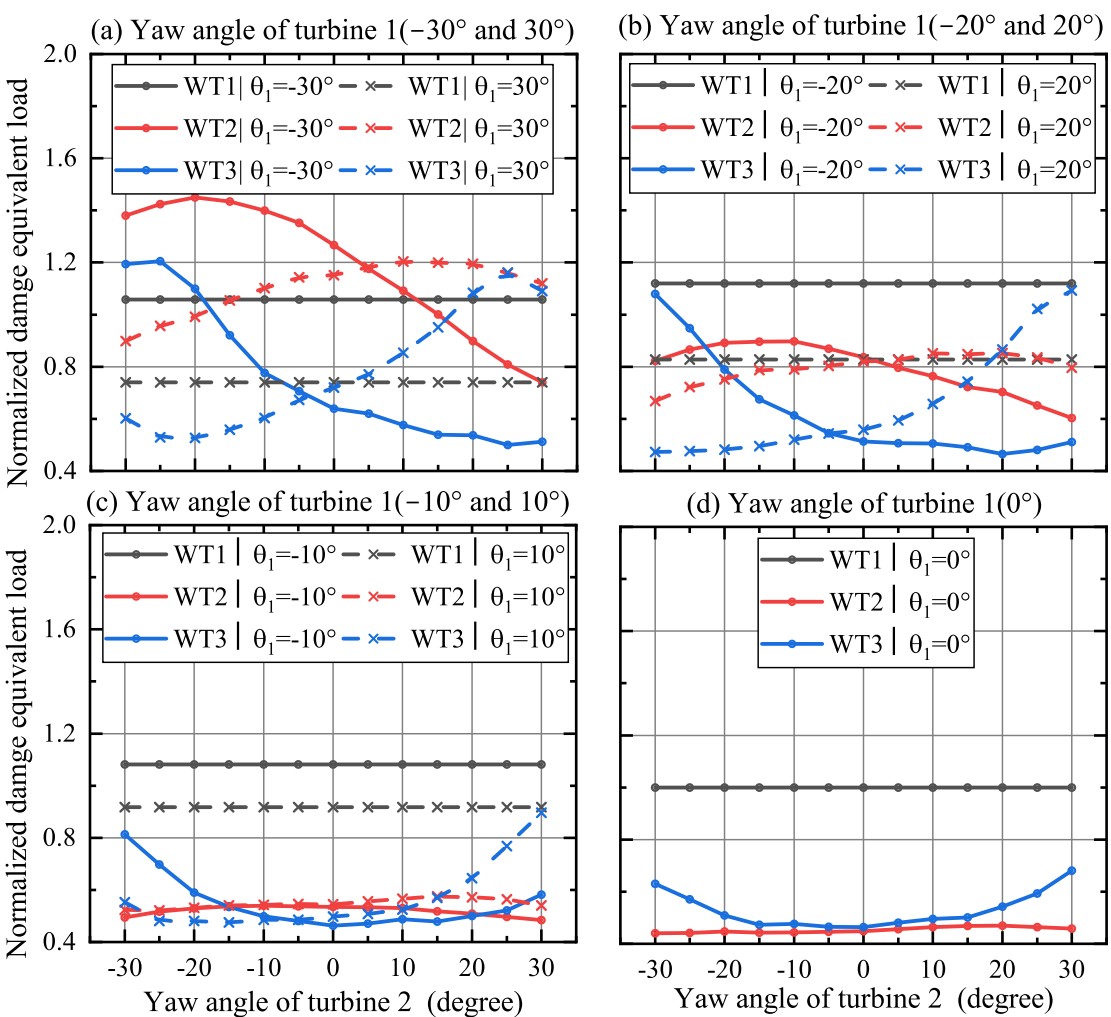

**Figure 12.** Blade root flapwise moment samples of normalized damage equivalent load under 9 m/s wind speed and 5% turbulence intensity. The damage equivalent load of turbine 1 under zero yaw offset case equals 1. The same axis scale is used in the four sub-pictures.

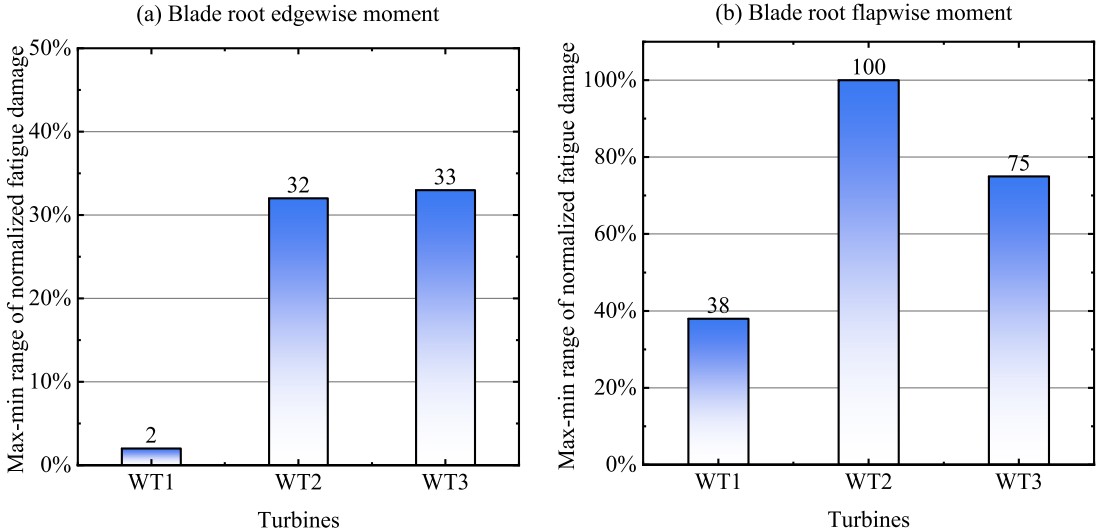

**Figure 13.** Max–min range of normalized fatigue damage of turbines.

Figure 14 presents the fatigue load of blade root edgewise moment under GP and LUT. As shown in Figure 14a,d,g, LUT predictions in WT1 are close to the samples. However, GP predictions are more partially fitted to the samples than LUT in Figure 14b,c,e,f,h,i. A similar trend can be seen in Figure 15.

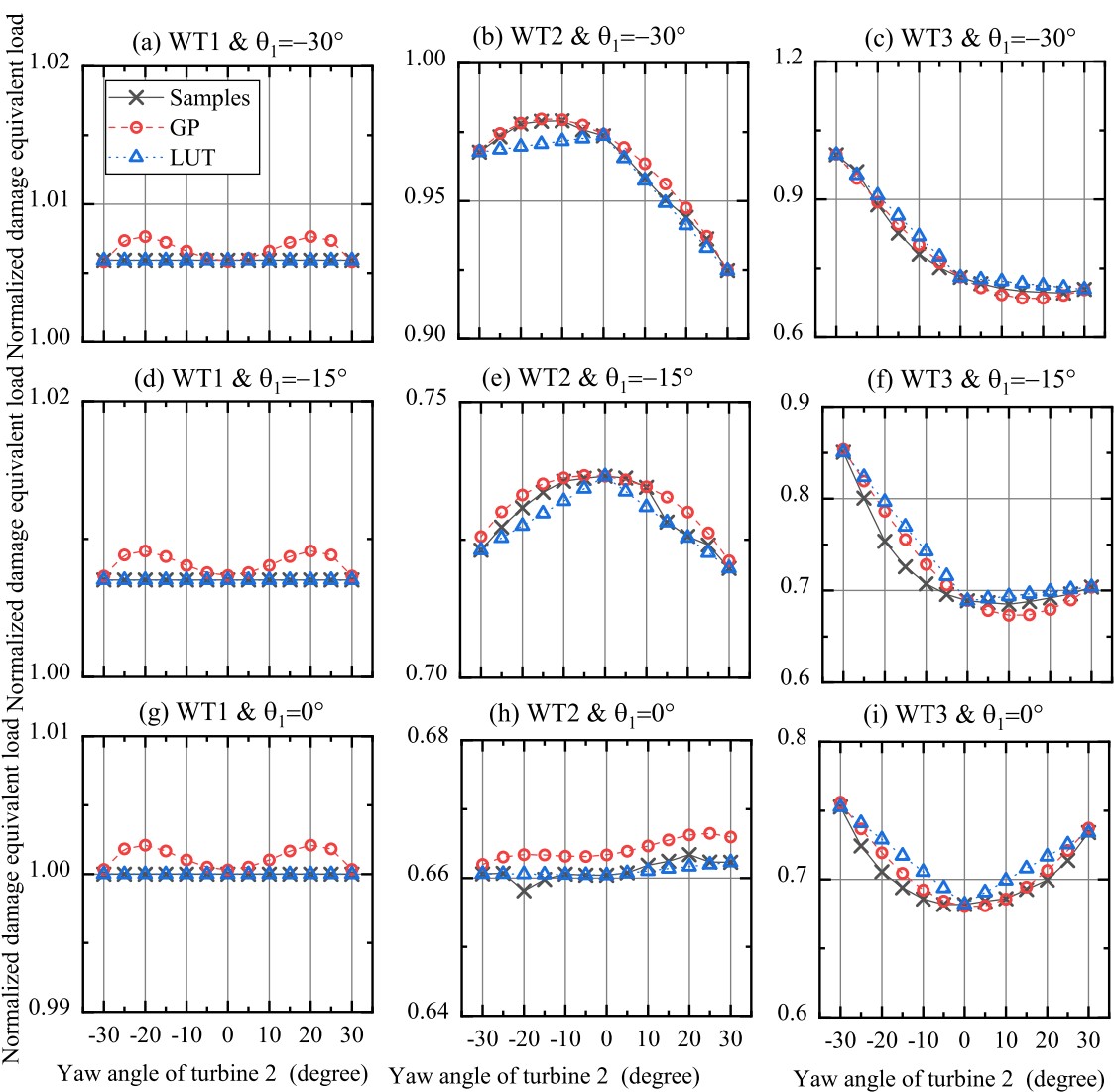

**Figure 14.** Prediction of normalized damage equivalent load at blade root edgewise moment under 9 m/s wind speed and 5% turbulence intensity.

**Remark 3.** *The higher the nonlinearity of the fatigue load, the smaller the GP prediction error compared to LUT.*

From the previous analysis, it was concluded that the wake of the upstream turbine influences the downstream turbine fatigue load and that the nonlinearity of this influence becomes stronger as the number of upstream turbines increases. Figures 16 and 17 show the errors of LUT and GP methods in three turbines.

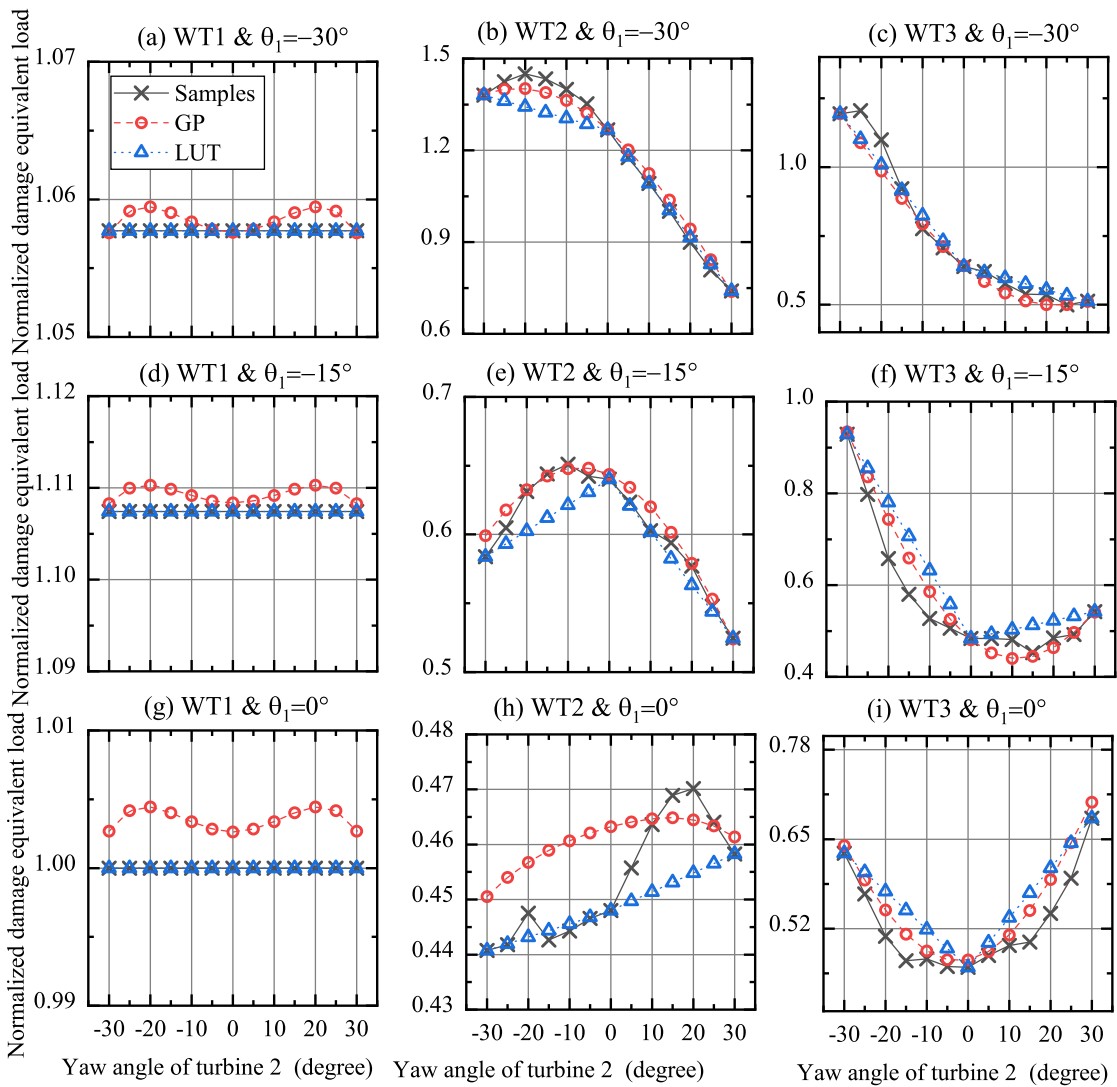

**Figure 15.** Prediction of normalized damage equivalent load at blade root flapwise moment under 9 m/s wind speed and 5% turbulence intensity.

Figures 16 and 17 show the errors of LUT and GP methods in three turbines. In both pictures, the prediction error increases from WT1 to WT3. Specifically, *RMSE* increased from 0 to 21.12% in Figures 16 and rose from 0 to 43.96% in Figures 17. A similar trend can be seen in *MAE* error. However, there are differences between the two pictures. For example, WT1 and WT2 have minor prediction errors lower than 1.2% on *RMSE* and 0.2% on *MAE* in Figures 16. In contrast, Figure 17 shows a more significant error of up to 22.46% on *RMSE* in WT2. The reason is that WT1 has weak nonlinear characteristics on BREM-DEL and BRFM-DEL, and WT2 only has weak nonlinear characteristics on BREM-DEL. LUT is more suitable for mapping the weak nonlinear characteristics between the fatigue load and yaw angles. For example, Figures 16 and 17 show that LUT has a zero error in WT1, compared to a slight error in GP, with nearly 1% *RMSE* and 0.1 *MAE*. Compared to LUT, GP is suitable for the nonlinear dataset. As shown in Figure 16, GP has a 5.18% *RMSE* and 0.99% *MAE* error, compared to 21.12% and 1.63% in LUT. Similarly, GP shows lower errors in Figure 17. The *RMSE* in GP is 6.99% (WT2) and 6.48% (WT3), compared to 22.46% (WT2) and 43.96%(WT3) in LUT. The same trend can be seen in *MAE*. In summary, the prediction accuracy of GP improved by 13.99% (*RMSE*) and 0.54% (*MAE*) at the blade root edgewise moment and 51.87% (*RMSE*) and 1.78% (*MAE*) at the blade root flapwise moment.

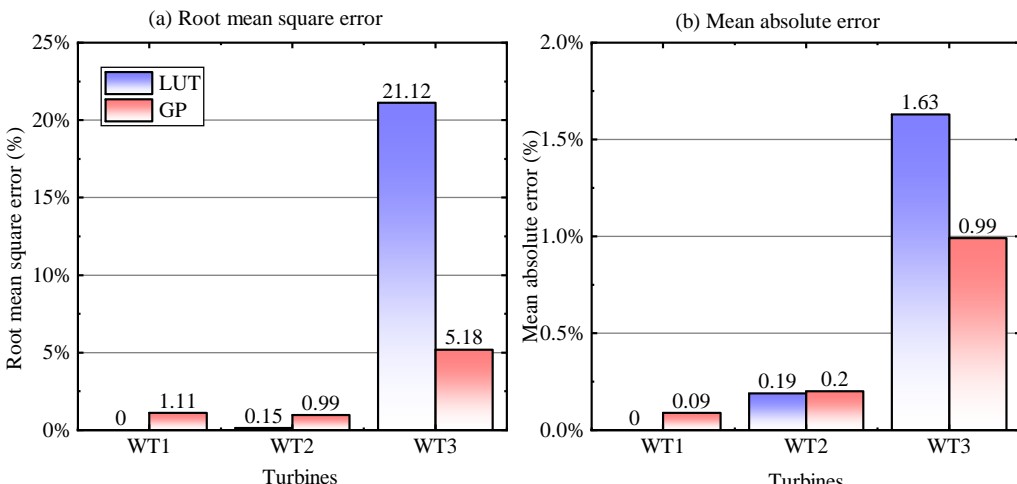

**Figure 16.** *RMSE* and *MAE* error of GP and LUT predictions on blade root edgewise moment.

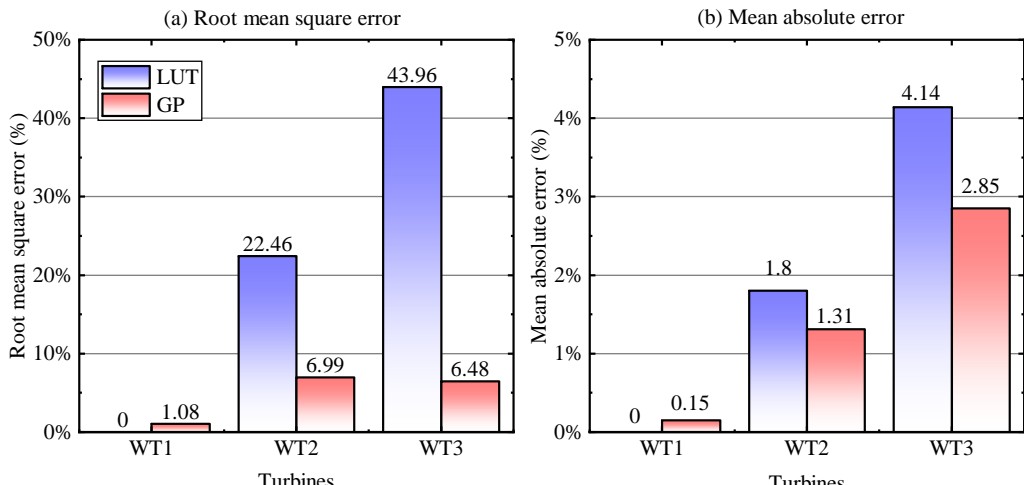

**Figure 17.** *RMSE* and *MAE* error of GP and LUT predictions on blade root flapwise moment.

## 5. Conclusions

This paper introduces a GP-based machine learning model to predict the damage equivalent load under yaw misalignment. This model considers the nonlinear relationship between fatigue load and yaw misalignment, which solves the problem of inaccurate prediction by LUT. The nonlinearity between the fatigue load and yaw misalignment strengthens with the increase of turbine depth (row number in the downstream direction); however, the sensitivity of the fatigue load to turbine depth is different. For example, in the three-turbine case, the damage of the blade root edgewise moment is very little affected by its yaw deflection, with a max–min range of 2% for the first row of turbines. However, as turbine depth increases, the max–min range increases to 32% (WT2) and 33% (WT3). The difference with the above load is that the blade root flapwise moment of the first row of turbines is influenced by 38%. As the turbine depth increases, the max–min range increases to 100% and 75%. Compared to the LUT algorithm, GP has a more accurate prediction of fatigue loads. For example, in the three-turbine case, the prediction accuracy of GP improved by 13.99% (*RMSE*) and 0.54% (*MAE*) at the blade root edgewise moment and 51.87% (*RMSE*) and 1.78% (*MAE*) at the blade root flapwise moment.

In the future, the wake steering control strategy will be optimized by farm-level fatigue load prediction and farm power prediction, thus enabling the wake steering control to balance power generation and turbine lifetime. Further, wake steering control will be combined with independent pitch control to reduce wind farm loads.

**Author Contributions:** Conceptualization, Y.M.; Funding acquisition, M.N.S.; Methodology, Y.M.; Supervision, A.H.; Writing—original draft, Y.M.; Writing—review & editing,Y.M., M.N.S. and A.H. All authors have read and agreed to the published version of the manuscript.

**Funding:** This research received no external funding.

**Institutional Review Board Statement:** Not applicable.

**Informed Consent Statement:** Not applicable.

**Data Availability Statement:** Not applicable.

**Conflicts of Interest:** The authors declare no conflict of interest.

## Abbreviations

The following abbreviations are used in this manuscript:

| | |
|---|---|
| WT | Wind turbine |
| WT1 | Wind turbine 1 |
| WT2 | Wind turbine 2 |
| WT3 | Wind turbine 3 |
| LUT | Look-up table |
| GP | Gaussion process |
| DELs | Damage equivalent loads |
| *RMSE* | Root-mean-square error |
| *MAE* | Mean absolute error |
| BREM | Blade root edgewise moment |
| BRFM | Blade root flapwise moment |

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
