# Peer review of "A Machine Learning Method for Modeling Wind Farm Fatigue Load"

_applsci, doi:10.3390/app12157392_

Round 1

Reviewer 1 Report

The manuscript entitled "A Machine Learning Method For Modeling Wind Farm Fatigue Load" has been investigated in detail. The topic addressed in the manuscript is potentially interesting and the manuscript contains some practical meanings, however, there are some issues which should be addressed by the authors:

1)  In the first place, I would encourage the authors to extend the abstract more with the key results. The "Abstract" section can be made much more impressive by highlighting your contributions. The contribution of the study should be explained simply and clearly.

2) The readability and presentation of the study should be further improved. The paper suffers from some minor language problems. The paper should be proofread by a native speaker or a proofreading agent.

3) Problem with citation (page 2, line 45) – no number provided

4) The Introduction section needs a major revision in terms of providing more accurate and informative literature review and the pros and cons of the available approaches and how the proposed method is different comparatively. Also, the motivation and contribution should be stated more clearly.

5) Figures are in general clear and informative.

6) Description of fig. 2 – maybe “Fatigue damage estimation method” ? Procedure?

7)  The method and performance of the proposed method should be better analyzed and discussed.

8) "Discussion" section should be edited in a more highlighting, argumentative way.

9) It will be helpful to the readers if some discussions about insight of the main results are added as Remarks.

10) Try to develop more conclusions form your research.

Author Response

Dear Editor and reviewers,

Best regards

Reviewer 2 Report

This manuscript proposes a machine-learning algorithm based on the Gaussian process to predict the fatigue load under yaw-based control that can improve the look-up table method's accuracy and overall power production of wind farms. The idea is good and sounded; however, many issues should be addressed according to the following comments:

1) The readability and presentation of this study must be further improved and simple to the reader. Further, more previous comparison results and discussions must be added. Please, correct the language problems, it is weak from the Grammarly and sequences of events, I catch 15 errors by using a personal program. The paper must be proofread very carefully by a native speaker or a proofreading agent.

2) The "Abstract" section should be more intensive and focused on the contribution of this manuscript directly supported with numerical results indicators. Also, the authors must not use the pronoun "we" in the whole manuscript.

3) The manuscript is too short, it is like a communication type not an article, please add more results, discussions, a flow chart, more information and analysis about the machine learning method (GP) supported with graphs of input datasets (training & testing ones).

4) The "Introduction" section should be made much more impressive and focused on the main idea directly by highlighting your contributions. The novelty of this manuscript must be explained simply and clearly in points at the end of the introduction section. Note that, the introduction section should consist of three parts, i.e., a general introduction to the topic, followed a literature survey, then the contribution clarifications.

5) The "Introduction" section should be enriched with up-to-date references by adding and citing the latest trends in the area of fatigue load prediction that can control and improve the overall power of wind turbines using machine learning techniques. E.g., Robust design of ANFIS-based blade pitch controller for wind energy conversion systems against wind speed fluctuations & Adaptive LFC incorporating modified virtual rotor to regulate frequency and tie-line power flow & Resilient design of robust multi-objectives PID controllers for automatic voltage regulators & Reliable Deep Learning and IoT-Based Monitoring System for Secure Computer Numerical Control Machines.

6) It is mandatory to check all the citing references of equations (1): (9). In addition, check carefully all the abbreviation definitions, symbols, and standard units in the whole manuscript according to the SI standard.

7) The resolution and quality of result figures should be modified; they should be presented as close to the camera-ready format. Also, please don't use the symbol abbreviations on X-Y-axes they must have the full name with their units. In addition, try to make zoom in at the effective area on each curve.

8) The conclusion section should be more concentrated and supported by the numerical results. Also, the authors may propose some interesting problems as future work in the conclusion.

Author Response

Dear Editor and reviewer,

Best regards

Round 2

Reviewer 2 Report

All of my concerns are adjusted, thanks.